# Epigenetic Aberrations in Major Psychiatric Diseases Related to Diet and Gut Microbiome Alterations

**DOI:** 10.3390/genes14071506

**Published:** 2023-07-24

**Authors:** Shabnam Nohesara, Hamid Mostafavi Abdolmaleky, Sam Thiagalingam

**Affiliations:** 1Department of Medicine (Biomedical Genetics), Boston University School of Medicine, Boston, MA 02218, USA; snohesar@bu.edu (S.N.); samthia@bu.edu (S.T.); 2Nutrition/Metabolism Laboratory, Beth Israel Deaconess Medical Center, Harvard Medical School, Boson, MA 02215, USA; 3Department of Pathology & Laboratory Medicine, Boston University School of Medicine, Boston, MA 02218, USA

**Keywords:** microbiome, nutrition, epigenetic, DNA methylation, histone acetylation, mental disease

## Abstract

Nutrition and metabolism modify epigenetic signatures like histone acetylation and DNA methylation. Histone acetylation and DNA methylation in the central nervous system (CNS) can be altered by bioactive nutrients and gut microbiome via the gut–brain axis, which in turn modulate neuronal activity and behavior. Notably, the gut microbiome, with more than 1000 bacterial species, collectively contains almost three million functional genes whose products interact with millions of human epigenetic marks and 30,000 genes in a dynamic manner. However, genetic makeup shapes gut microbiome composition, food/nutrient metabolism, and epigenetic landscape, as well. Here, we first discuss the effect of changes in the microbial structure and composition in shaping specific epigenetic alterations in the brain and their role in the onset and progression of major mental disorders. Afterward, potential interactions among maternal diet/environmental factors, nutrition, and gastrointestinal microbiome, and their roles in accelerating or delaying the onset of severe mental illnesses via epigenetic changes will be discussed. We also provide an overview of the association between the gut microbiome, oxidative stress, and inflammation through epigenetic mechanisms. Finally, we present some underlying mechanisms involved in mediating the influence of the gut microbiome and probiotics on mental health via epigenetic modifications.

## 1. Introduction

Mental disorders are known as global and serious problems which influence one in four people during their lives, and both genetic and environmental factors play an important role in their disease pathogenesis [1,2]. Although genes are essential in developmental processes and in maintaining the anatomical structure and functionality of any species, large-scale genetic studies in mental diseases have been unable to identify specific genes or several genes responsible for the pathogenesis of mental diseases. Instead, several hundred genetic mutations were found to be involved in the pathogenesis of any specific major mental disease, but the effect size of none of them reached even 1% [3,4]. Hence, it has been proposed that environmental factors through epigenetic alterations may play more significant roles in the pathogenesis of mental diseases [5,6]. Among various environmental factors such as toxins, contaminants, and malnutrition, the potential impacts of gut microbiota which affect and are affected by nutritional components have been under extensive scrutiny in recent years. Notably, every year the human body loses almost its entire body composition (except neuronal cells) by weight due to apoptosis and cell death, but regenerates it as instructed by genetic and epigenetic codes and using nutritional elements consumed during that year [7]. Surely any aberration in applying genetic or epigenetic instructions due to interfering environmental factors such as toxins, contaminants, and age-related de novo mutations as well as malnutrition (shortage or surplus of specific nutritional elements) could derail the structural and functional (re)formation of the regenerated cell. Considering that genetic codes are relatively stable throughout life, although each brain cell may contain more than 1000 de novo genetic mutations [8], the main focus of this review article is on the epigenetic landscape which is dynamic and more susceptible to environmental factors, including diverse products of the gut microbiome, diet composition and nutritional elements, which interactively may affect epigenetic marks linked to mental diseases.

Epigenetics refers to modifications to DNA or RNA capable of influencing gene expression levels without alerting the underlying DNA sequence [9]. During the last two decades, it has been well documented that the derangement of epigenetic mechanisms and their molecular machinery is associated with the onset and progression of several mental disorders [10]. On the other hand, several lines of experimental evidence indicate that environmental factors like lifestyle and nutritional deficiency or surplus as well as internal microenvironments in our bodies such as gut microbiome (the internal ecosystem) change epigenetic codes [11,12].

As more than 1000 bacterial elements (predominantly nonpathogenic bacteria) as well as viruses, archaea, protozoa, and fungi live in a healthy intestinal ecosystem, they display powerful roles in metabolic functions, immunomodulation, and modulating gene expression via epigenetic mechanisms [13]. Microbiomes are categorized into several types including symbiotic, commensal, and pathogenic or dysbiosis types, and modify the effects of diet/nutrition in humans and other multicellular organisms with digestive tracts [14]. It had been estimated that the total population of gut bacteria is ten times more than the total number of human body cells [15]. However, a more recent estimate indicates this ratio as 1:1 [16]. Interestingly, the collective genes of human microbiota far exceed the number of human genes (almost 100-fold); 3.3 million non-redundant microbiota genes versus 25,000–30,000 in humans [15]. As gut microbiota and human genes have a mutual relationship and co-evolved, and considering their huge cell count, large number of genes, and their functional capacity within the human body (Figure 1), the gut microbiota is considered an important “virtual” or “metabolic organ” [17].

The intestinal microbiome is involved in numerous physiological processes including host defense [18], homeostasis [19], immunomodulation [20], metabolism of amino acids and glutathione [21,22], detoxification of dangerous substances [23], synthesis of vitamins and extra advantageous metabolites [24], mediating digestion of non-digestible dietary fiber and producing short-chain fatty acids (SCFA) [25].

An imbalance of the gut microbiome occurs when people are subjected to aggressive factors including an unhealthy diet [26,27], stress [28], antibiotic use [29], smoking [30], and excessive consumption of alcohol [31] which in turn leads to the decrease of commensal bacteria, the increase of pathogenic bacteria, and microbiome dysbiosis [32]. Microbiome dysbiosis might trigger a sequential chain of events including disruption of metabolic processing, epigenetic changes, enhancing oxidative stress, inflammation, onset and progression of major mental disorders like schizophrenia, autism spectrum disorder (ASD), bipolar disorder, and major depressive disorder (MDD) [33,34,35]. Here, we probe the effect of alterations in the microbial structure and composition in shaping specific epigenetic alterations in the brain and their role in the onset and progression of major mental diseases. We will also provide an overview of potential interactions among maternal diet/environmental factors, gastrointestinal microbiome, and accelerating or delaying the onset of mental disorders via epigenetic changes.

## 2. Epigenetic Aberrations and Mental Disorders

Epigenetic alterations play significant roles in the pathogenesis of severe mental disorders like schizophrenia, autism, bipolar disorder, and depression [36,37,38]. Epigenetic regulations involving the nucleosome include DNA methylation, and histone modifications [39,40,41]. Silencing genes by DNA methylation occurs based on two principal mechanisms. First, methylation in critical sites confers suppressing the binding of transcription factors to their recognition elements [42,43]. Second, the methylation of a regulatory region of DNA mediates the recruitment of chromatin modification enzymes and methylated DNA binding proteins to the gene regulatory regions for facilitating or silencing chromatin via introducing histone modifications [44,45].

Epigenetic mechanisms have been found to be flexible which can be changed by a variety of stimuli like stress and environmental conditions [46,47]. Emerging evidence has demonstrated that epigenetic mechanisms are key factors in immunological memory and mental disorders [48,49]. Stable epigenetic variations in the innate and adaptive immune cells have been found in people subjected to early-life stress which in turn accelerates the development of stress-induced mental disorders [50]. Regarding specific mental disease, a highly significant correlation between methylation profiles in the blood of 581 MDD patients at baseline and future disease status (6 years later) has been reported [51]. The pathogenesis of schizophrenia and bipolar disorder is also correlated with hypomethylation of the A3 CpG site in both diseases and hypermethylation of the AluY A1 and A2 CpG sites in bipolar disorder [52]. In addition, a methylome-wide analysis of DNA methylation of peripheral blood samples from 106 Chinese schizophrenia family trios, 125 hypomethylated and 112 hypermethylated regions were detected [53]. Another study demonstrated that the development of schizophrenia and intellectual disability is associated with the altered methylation levels of FYN, signal transducer and activator of transcription 3 (STAT3), Ras-related C3 botulinum toxin substrate 1 (Rac1), and nuclear receptor 4A2 (NR4A2) as well as changing function of the immune system [54]. More examples of diverse epigenetic alterations in major mental diseases have been presented elsewhere [5].

Other studies have shown that DNA methylation and its physiological effects on the host can be regulated by commensal bacteria in colonic epithelial cells (CoECs) [55], one of the main elements of gut-blood connections. Commensal bacteria have also demonstrated abilities for controlling the expression of Aldh1a1, which is responsible for encoding a retinoic acid-producing enzyme and plays a powerful role in the maintenance of intestinal homeostasis through DNA methylation in the overlapping 5′ region of Tmem267 and 3110070M22Rik genes in CoECs [56].

Besides DNA methylation, diverse types of histone modifications (e.g., histone acetylation and methylation) are among other important epigenetic mechanisms for gene expression regulation affected in mental diseases. Histone acetyltransferases (HATs) mediate acetylation of specific lysine residues on the N-terminal tail of core histones which in turn give rise to DNA uncoiling and increasing accessibility to transcription factor binding. Conversely, histone deacetylases mediate chromatin remodeling that is involved in suppressing gene transcription [57]. Therefore, histone deacetylase inhibitors such as SCFAs (mostly butyrate, propionate, and acetate which are also produced by gut microbiota) have garnered much attention from the scientific community as good candidates for the treatment of mental disorders [58]. Considering the examples mentioned above supporting the powerful roles of epigenetic alterations in the pathogenesis of mental disorders like schizophrenia, it can be concluded that epigenetic alterations are among the key therapeutic targets to improve mental health.

## 3. Microbiome–Gut–Brain Axis Influence on Brain Functions Mediated by Epigenetic Modification

The interplay between the intestinal microbiome and the brain through the gut–brain axis, results in extra-intestinal effects [59]. The gut and the brain are linked together via the Vagus nerve which creates a connection from the enteric nervous system (ENS) to the CNS [60]. Signal transmission from the gut microbiome to the CNS and ENS can be mediated via different neuronal, immune-mediated, and metabolite-mediated pathways [61,62]. Accumulating evidence has shown that gut microbes can be considered master regulators of the CNS function by producing microbial metabolites, regulating the Vagus nerve, promoting the generation of neurotransmitters, the activation of the immune system, and the mediation of inflammation (Figure 2) [63,64]. The compounds of microbial origin can be sensed by chemoreceptors present on the Vagus nerve [65], but the gut–brain axis is influenced by the inflammatory cytokines and antimicrobial peptides secreted by microbes as well [66]. Nevertheless, the gut microbiome is also involved in the synthesis of vitamins, and neurotransmitters, producing the SCFAs (acetate, propionate, and n-butyrate), energy harvesting from food, and the absorption of nutrients that might cause epigenetic alterations [67,68]. Some members of the gut microbiota like the Lachnospiraceae family are responsible for the breaking down of carbohydrates into SCFAs such as butyrate, propionic acid, acetic acid, and valeric acid [69]. SCFAs have been found to be the functional and critical components in the microbiome–gut–brain axis and are used as almost 10% of the human energy source [70]. Changes in microbiome diversity and composition can result in derangement in SCFA production and subsequently dysfunction of the intestinal barrier and the host brain development [71,72].

Microbiome-derived SCFAs also act as histone deacetylase inhibitors and play important roles in numerous physiological processes such as promoting the differentiation of T cells, producing neurotransmitters, regulating immune homeostasis, inhibiting intestinal pro-inflammatory macrophage function, regulating gut and blood–brain barrier (BBB) permeability, and neuroprotection [73,74,75,76,77]. Therefore, a variety of mechanisms such as DNA methylation, histone acetylation or methylation and other types of histone signatures, and chromatin plasticity might be affected by microbiome and gut microbiome-derived metabolites [33]. For example, the afferent fibers on the Vagus nerve are directly activated by butyrate [78]. SCFAs are produced by the fermentation of indigestible polysaccharides in the diet and can activate vagal afferents through free fatty acid receptors to send signals to the brain. They also have demonstrated abilities to cross the BBB through monocarboxylic acid transporters present on the endothelial cells (ECs). The integrity of the BBB is affected by SCFAs since they suppress inflammation-related pathways and contribute to modulating neuroinflammation by tailoring the microglia morphology and function [79]. SCFAs are involved in other processes in the brain such as promoting the synthesis of 5-hydroxytryptamine (5HT) via histone acetylation, regulating levels of neurotrophic factors in the CNS, emotion, learning, and memory [80]. Reprogramming of the cellular epigenome, both proximally and distally, can be also mediated by gut microbiome-derived metabolites such as the histone deacetylase inhibitor butyrate. Furthermore, butyrate contributes to mitochondria-located sirtuins-3 and enhances the deacetylation and disinhibition of the pyruvate dehydrogenase complex which in turn gives rise to better optimized mitochondrial function in reactive cells, central glia, and systemic immune cells [81]. Considering specific mental disorders, abnormal serum levels of SCFAs have been found in the early phase of schizophrenia [82]. Another recent clinical study showed that alterations in concentrations of SCFAs can be linked to the pathogenesis and the cognitive impairment of schizophrenia. Additionally, a significantly lower concentration of valeric acid was reported in schizophrenia patients compared to healthy controls [83]. According to the “membrane hypothesis of schizophrenia”, the abnormal production of SCFAs occurring by the altered composition of the gut microbiome may result in the activation of microglia and then disrupt cell membrane metabolism [84].

Regarding methylation reactions, although the availability of methyl groups for DNA or histone methylation in the CNS and consequently neurobehavioral functions can be affected by nutritional status, it is also regulated by the host’s microbiome. For example, as choline (found in vegetables like spinach) is the main source of methyl groups, some bacteria compete with the host and consume this nutrient intensifying metabolic diseases associated with global DNA methylation alterations in mice under high-fat diet and in their offspring [85]. Methionine, an essential amino acid (and the main methyl donor), is metabolized by several other bacteria as a nutrient source, while other bacteria such as *Escherichia coli* can produce methionine through specific enzymatic pathways [86]. Production and consumption of folic acid as well as vitamin B-12 both of which play significant roles in methylation reactions (as the carriers of methyl groups) are under the influence of gut microbiome as well [87].

A variety of serious diseases especially mental disorders are linked to differences in gut microbiome composition and function [88,89]. For example, a significantly reduced abundance of the *phylum Bacteroidetes* and an increased abundance of the *Actinobacteria* and *Firmicutes* have been found in bipolar disorder with current major depressive episodes [90]. However, their therapeutics may also alter gut microbiome composition. For instance, atypical antipsychotic (AAP) treatment in patients with bipolar disorder or schizophrenia resulted in measurable differences in the gut microbiome, especially increased abundance of the *Actinobacteria phylum* [91].

Table 1 shows some other studies in which differences in the microbial structure and composition result in changes in producing metabolites and bacterial substances and consequently the onset and development of mental disorders via epigenetic alterations. In summary, as differences in microbial structure and composition are linked to alterations in producing vitamins, metabolites, and bacterial substances, they are involved in the onset and progression of mental disorders through epigenetic changes [92].

## 4. Maternal Diet, Offspring Gut Microbiome, and Brain Functions

There is a microbial connection between mother and offspring from the early moments of life [107]. The changes in the mother’s vaginal, oral cavity, and stomach microbiome have been reported throughout pregnancy [108]. There are numerous factors such as nutrition, psychological tensions, host genomes, pathogens, and medications that affect maternal-infant microbial characteristics and composition [109]. The maternal milk nutrients and bioactive components along with the maternal microbiome are responsible for the early establishment and maintenance of the infant gut microbiome. The infants’ gut microbiome is also influenced by paternal and other postnatal factors such as geographic location and seasonal of birth which affect exposure to natural light, thus circadian rhythm and maternal dietary patterns and food availability. Accordingly, as illustrated in Figure 3, growing bodies of evidence have shown that these environmental factors and maternal diet strongly affect offspring brain development predisposing to mental disorders by changing microbiome composition and epigenetic mechanisms [110,111,112,113]. Although bioactive food components and dietary nutrient intake are considered crucial environmental factors that affect histone acetylation or methylation and DNA methylation, their effects are implemented either via suppressing enzymes involved in catalyzing DNA methylation and histone acetylation/methylation or through altering the availability of essential substrates for those enzymatic reactions which utilize acetyl or methyl groups [114].

For example, acetate production from the breakdown of alcohol gives rise to dynamic acetylation of histones in the brain tissue. In a pregnant mouse, the incorporation of labeled acetyl groups into gestating fetal brains occurred following exposure to labeled alcohol. Extracellular acetate could mediate transcriptional programs relevant to learning and memory, highlighting the importance of gut microbial metabolites in epigenetic regulating learning and memory [115]. Additionally, as non-absorbable antibiotics lead to gut microbiome depletion, there is a report on subsequently reduced locomotor sensitization to morphine, along with noticeable changes in inflammatory gene expression, neuronal functions, and behavior [116]. Chen et al. also found that the behavioral changes in methamphetamine-exposed mice are associated with the reduction in SCFAs that is linked to inflammation and epigenetic modification [117]. Bisphenol A (BPA) is considered to be another environmental factor (in fact a contaminant) that may affect the neurobehavioral system during childhood and adulthood periods owing to its extensive use in the production of polycarbonate plastics and epoxy resins such as food or water containers, dental sealants, and thermal papers [118,119,120,121]. Although BPA is known to reduce global DNA methylation [122], altering neurogenesis following in vivo maternal and in vitro BPA exposure has been attributed to an increase in the epigenetic regulator lysine-specific histone demethylase1 (LSD1), as well [123]. It was also shown that dietary intake of BPA for 22 weeks could impair learning and memory in male mice by enhancing neuroinflammation and disruption of the BBB along with altering the diversity and composition of the gut microbiome, functional profile changes in the microbial community and reduced levels of neurotransmitters (tryptophan), 5-hydroxytryptamine (5-HT), and 5-hydroxyindoleacetic acid (5-HIAA) in colon, serum, and hippocampus region [124]. In light of these findings, one can conclude that both maternal diet and environmental factors during pregnancy profoundly influence the diversity and composition of the gut microbiome which subsequently play significant roles in accelerating or delaying the onset of severe mental illnesses in offspring via epigenetic alterations.

Figure 3 illustrates the maternal diet and environmental factors that affect the gut microbiome and epigenetic programing in offspring and subsequently influence brain functions and cognition.

Table 2 shows a summary of the main findings that address potential interactions among maternal diet/environmental factors and gastrointestinal microbiome, which accelerate or delay the onset of mental disorders via epigenetic changes.

## 5. Gut Microbiome, Oxidative Stress, Inflammation, and Epigenetic Changes

Gene regulation and expression can be tuned by gut microbial products because gut bacterial-derived metabolites act as epigenetic agents. It has also been reported that host molecular signaling can be influenced by gut bacterial-derived metabolites. Physiological effects of SCFAs are mediated via at least two mechanisms: (i) they are capable of inhibiting histone deacetylases through acetylation of lysine residues which in turn confers promoting the binding of transcription factors to genes promoter regions for regulating gene expression [144]; (ii) they can bind to G-protein-coupled receptors (GPCRs) regulating energy metabolism and immune response through their activation and inhibiting histone deacetylases (HDACs) [73,145]. In addition, gut microbial products may mediate or even substitute host reactive oxygen species (ROS) production. ROS has been found to be the second messenger that alters inflammatory, immune, and other signaling processes by exerting oxidative activity on proteins. Moreover, interactive effects of ROS and epigenetic mechanisms have been reported. For example, ROS participates in histone/protein deacetylation, chromatin remodeling, and the activation of transcription factors [146]. Collectively, these lines of evidence indicate that an increase in ROS levels and inflammation by some gut bacteria metabolites are linked to dysregulation of the gut–brain axis leading to the pathogenesis of mental disorders providing greater insights to improve mental health by clarifying their underlying mechanisms.

## 6. Roles of Gut-Blood Barrier, Microbiome-Derived Metabolites, and Diets in the Progression or Therapy of Mental Disorders

The preservation of intercellular junctions, epithelial cell architecture, and the integrity of the epithelial barrier heavily depend on the gut microbiome. The leaky gut is defined as the lack of integrity of the epithelial barrier and increasing the permeability of the intestinal mucosa, which in turn can result in the development of many systemic, intestinal, and even brain dysfunctions. The leaky gut provides the opportunity for bacteria, bacterial toxins, toxic digestive metabolites, and small molecules to translocate systemically, and even cross the BBB and induce neuroinflammation [147]. For instance, Maes et al. reported that increased bacterial translocation is associated with the pathophysiology of chronic depression and immune responses [148]. They suggested two mechanisms in which bacterial translocation confers systemic inflammation in depression: (i) it can be considered a primary trigger factor for the onset of depression in some people and, (ii) it might have taken place secondary to systemic inflammation and exacerbate neuroinflammation in depression. Nevertheless, a healthy dietary and nutrient composition strongly influences the gut microbiome epithelial barrier and subsequently can inhibit the onset and progression of mental disorders [149]. For example, the Mediterranean diet (MD) has been known to be an ancient dietary pattern with beneficial health effects [150] which tunes microbial diversity and microbiome composition [151,152,153]. Interestingly, alpha diversity assessment using the Simpson index showed that MD is capable of increasing the bacterial diversity of five bacterial genera compared to a “typical” American diet in just 2 weeks [154]. Solch and coworkers reported that MD could also improve cognitive function by creating beneficial changes in the gut microbiome, decreasing the abundances of *Bifidobacterium* and *Erysipelatoclostridium* and increasing the abundance of six genera such as *Candidatus Saccharimonas*, *Lachnoclostridium*, and *Romboutsia*, compared to the Western diet [155]. Experimental evidence also indicates that the use of MD during pregnancy may hamper the offspring’s disease development via epigenetic alterations [156]. For example, maternal adherence to MD in early pregnancy resulted in favorable neurobehavioral outcomes in early childhood, mainly due to methylation differences for regulatory regions of imprinted genes; paternally expressed gene 10 (PEG10)/(epsilon (ε)-sarcoglycan SGCE), maternally expressed gene 3 (MEG3) and insulin-like growth factor 2 (IGF2) [157]. In another study, a total of 1573 differentially methylated regions were identified in people with polyphenol-rich green MD compared to individuals using a normal diet (based on a healthy dietary guideline) which was linked to serum-folic acid levels [157].

On the other hand, a high-fat diet (HFD) can lead to increased induction of DNA damage, higher inflammatory activities, and altered gene expression and DNA methylation of DNA methyltransferases1 (DNMT1) and MutL homolog 1 (MLH1) genes, which are involved in DNA methylation and mismatch repair, respectively, but (-)-epigallocatechin gallate (EGCG), a component of tea can attenuate these alterations [158]. Kim et al. reported that the excessive intake of an HFD could cause disturbance of gut microbiota composition, especially an increase in the Proteobacteria population, which in turn resulted in increasing plasma concentration of lipopolysaccharide (LPS), inhibiting hippocampal brain-derived neurotrophic factor (BDNF) expression, and consequently inducing the phenotypes of psychiatric disorders in mice [159]. In another study, Wang et al. found that there are close relationships between depressive and anxiety-like behaviors in long-term multiple nonsocial stress-treated mice and stress-induced gut barrier integrity damage and enhanced LPS levels in both serum and brain cortex. In their study, sesamin, a dietary fat-reduction supplement, could inhibit both colon and cortex inflammatory responses by suppressing tumor necrosis factor alpha (TNFα) and interleukin 6 (IL-6) expression [160]. However, the effects of HFD on the gut microbiome in overweight mice with mood ailments are strongly dependent on sex. For instance, reducing locomotion activity in response to stress and vulnerability to the anxiogenic effects of the high-fat diet is higher in male than female mice [161].

In humans, the lower levels of paraoxonase 1 enzyme (PON1) activity in schizophrenia, leading to impaired function of the innate immune system and increased oxidative stress, has been linked to increased IgA responses to *Pseudomonas putida*, *Morganella morganii*, and *Pseudomonas aeruginosa* [162]. With the involvement of gut microbiome in disease pathogenesis, it is not surprising that differential DNA methylation sites in schizophrenia are enriched in genes related to maturation and activation of immune cells, associated with increased levels of autoantibodies [163]. In this line, there is a similar report indicating widespread alterations in methylation levels of genes related to immune cell activity in 39.7% of schizophrenia patients, accompanied by a lower proportion of lymphocytes and a higher proportion of neutrophils [164].

### 6.1. Gut Microbiome-Derived Metabolites May Alleviate Mental Disorders via Epigenetic Alterations

Although the gut microbiota composition has been found to be a critical factor in modulating the bioavailability of phenolic acids (present in the total dietary polyphenolic food) associated with memory promotion [165], it has been shown that microbiome-derived phytochemicals can act as epigenetic modifiers to reduce immune inflammatory responses in MDD. In an interesting study, Pasinetti et al. examined the efficacy of two novel phytochemicals (dihydrocaffeic acid (DHCA) and malvidin-3′-O-glucoside (Mal-gluc)) derived from post-absorptive and microbiome metabolism of bioactive dietary polyphenol preparation (BDPP) in reducing depression via epigenetic modulation [166]. It was also shown that DHCA could attenuate the production of pro-inflammatory cytokines such as IL-6 by inhibiting DNA methylation at the CpG-rich IL-6 sequences in introns 1 and 3. In addition, Mal-gluc could contribute to modulating synaptic plasticity via enhancing histone acetylation of the regulatory sequences of the Rac1 gene.

Furthermore, the resident gut bacteria have demonstrated abilities for the fermentation of dietary fiber and consequently for producing SCFAs which are mediators of improving mood and cognition, and reducing the risk of psychiatric diseases like depression in humans [167]. The gut microbiome and its resultant metabolome are also known to be master regulators of transcriptional changes in the frontal cortex and subsequently behavioral changes such as emotional, social, and cognitive functions [168]. As epigenetic changes in the gut are mainly mediated by direct effects of the gut microbiome, epigenetic changes in the brain can be induced by their fermentation products. In other human studies, while abnormal serum levels of SCFAs have been reported in the early phase of schizophrenia and ASD that are associated with cognitive dysfunction [82,104], some previous reports demonstrated that in addition to diet which is a crucial factor in the modification of histone acetylation, SCFAs or other gut microbiome-derived metabolites are responsible for such alterations [169,170].

Acetate, another microbial metabolite is known to be an important epigenetic regulator since it can inhibit histone deacetylase activity and change gene expression and behavioral responses in mice brains [115,171]. Therefore, the use of microbiome-related metabolites such as acetate could be a reasonable approach to reverse social deficits in ASD via transcriptional changes in the frontal cortex. To this end, Kiraly et al. examined the neuroprotective effects of acetate supplementation in mutant mice carrying a deletion of the ASD-associated Shank3 gene (Shank3^KO^) [172]. In addition to social deficits, functional and compositional changes in the gut microbiome were seen in Shank3^KO^ mice. Their results revealed that supplementation with the microbial metabolite acetate could hamper social deficits in microbiome-depleted mice by altering transcriptional regulation in the prefrontal cortex. Olaniyi and coworkers also showed that acetate could rescue polycystic ovary syndrome (PCOS) associated depression by inhibiting DNA methyltransferase in prefrontal and hippocampal regions of the brain in adult female Wistar rats [173].

Butyrate is another SCFA and a secondary metabolite of the gut microbiome which acts as an endogenous inhibitor of class I HDACs, a well-known class of epigenetic modulatory genes [174,175]. Neuroprotective effects of butyrate against ASD have been linked to positive modulation of mitochondrial functions like enhancing oxidative phosphorylation and beta-oxidation, suppressing microglia-mediated neuroinflammation, regulating the microbiome–gut–brain axis, reducing oxidative stress-induced transport deficits of tryptophan or its activity as a histone deacetylase inhibitor [176,177,178,179]. Moreover, butyrate as an HDAC inhibitor can enhance acetylation around the promoters of neurotrophic factors such as BDNF and consequently promote their transcription [180,181].

A preclinical study revealed that sodium butyrate could ameliorate social behavior deficits by modifying the transcription of inhibitory/excitatory genes in the frontal cortex of the BTBR autism mouse model [182]. Moreover, Xu et al. reported reducing histone3-lysine9-β-hydroxybutyrylation (H3k9bhb), a novel histone modification mark, in the brain of depressive mice model and found that exogenous β-hydroxybutyrate could attenuate depressive behaviors by restoring H3K9bhb and BDNF [183].

Indole 3-propionic acid, an enteric microbiome-derived deamination product of tryptophan also contributes to intermittent fasting-dependent axonal regeneration by modulating the function of neutrophils and interferon-gamma [184,185]. In another study, it has been reported that 3(3,4-dihydroxy-phenyl) propionic acid, as a novel microbiome-derived epigenetic modifier, could reduce IL-6 expression in human peripheral blood mononuclear cells (PBMCs) via regulating methylation of CpG motifs in its promoter region. Figure 4 illustrates the cascades of events that gut food-microbiome products influence blood cells and other related tissues and finally affect brain functions and cognition.

### 6.2. Ketogenic Diet for the Treatment of Mental Disorders via Epigenetic Changes

The potential health benefits of a ketogenic diet and its underlying molecular mechanisms can be mediated through epigenetic mechanisms such as elevating histone acetylation [186,187,188]. In addition, the protective effects of ketogenic diets against mental disorders such as ASD can be due to modification of the gut microbiome and its secondary metabolites [189,190,191].

Some previous reports have shown that a ketogenic diet can be considered to be an effective remedy to rescue social interaction deficits in a Shank3 mouse model of autism because β-hydroxybutyrate, a secondary metabolite of gut microbiome and an endogenous inhibitor of class I HDACs can be produced by this diet. In an interesting study, Yan et al. reported reduced levels of the HDACs downstream target genes encoding the N-methyl-D-aspartate (NMDA) receptor subunits, glutamate ionotropic receptor NMDA type subunit 2A (GRIN2A) and glutamate ionotropic receptor NMDA type subunit 2 B (GRIN2 B) in the prefrontal cortex of an autism mouse model with a deficient Shank3 gene [192]. It was also shown that the ketogenic diet can improve social behavior deficits through the restoration of histone acetylation and gene expression in the prefrontal cortex, which was attributed to β-hydroxybutyrate generation by the ketogenic diet. Another study using a mouse model of idiopathic ASD reported that a ketogenic diet could reduce oxidative stress (by altering lipid peroxidation levels) and secretion of inflammatory cytokines, and increase the relative abundance of putatively beneficial microbiota (e.g., *Akkermansia* and *Blautia*) [193].

### 6.3. Probiotic Therapy of Mental Disorders via Epigenetic Changes

As fecal transplantation from patients with autism, schizophrenia, and MDD to germ-free mice induces corresponding disease-like phenotypes associated with related neurochemical and metabolic alterations in the recipient mice, probiotics were shown to be useful in reversing social and cognition deficits by targeting the gut–brain axis [194,195]. The main mechanisms of action of probiotics in reducing social cognition deficits in mental disorders are epigenetic alterations, immuno-modulatory properties, promoting biosynthesis of neurotransmitters, the anti-inflammatory antioxidant potential, and downstream regulation of the hypothalamus-pituitary-adrenal (HPA) axis [196,197]. In a study by Wang et al. a greater *Turicibacter* abundance and lower butyric acid levels were reported in a valproic acid (VPA)-induced autism mouse model, and the treatment with *Lactobacillus* could alleviate autistic-like behaviors by reducing *Turicibacter* abundance and elevating butyric acid levels [198]. Probiotic therapy is also capable of alleviating memory dysfunction by targeting epigenetic events. In an interesting study by Wang et al., a long-term probiotic supplement resulted in the normalization of the gut microbiome composition, alleviating memory dysfunction in lead-exposed rats by restoring the reduced H3K27me3 (trimethylation of histone H3 Lys 27, an epigenetic mark mediated by EZH2) in the hippocampus of adult rats [199]. In another study, it was shown that *L. reuteri PBS072* and *B. breve BB077* are excellent probiotic candidates for improving cognitive functions and stress resilience via suppressing the epigenetic enzyme LSD1, increasing gamma-aminobutyric acid (GABA), and serotonin [200]. Additionally, Duan et al. reported the anti-neuroinflammatory effect of the potential probiotic *Roseburia hominis* (*R. hominis*) and its possible molecular mechanisms in germ-free rats [201]. In their study, *R. hominis*-treated group could exhibit increased serum levels of propionate and butyrate (as histone deacetylase inhibitors), reduced levels of monocyte chemoattractant protein-1 (MCP-1), interferon-gamma (IFN-γ), and interleukin-1 alpha (IL-1α), and reducing microglial activation compared to germ-free rats. Therefore, gut microbiome-derived metabolites, ketogenic diet, and probiotic therapy can be considered promising remedies to alleviate memory dysfunction and social cognitive deficits in mental disorders by targeting epigenetic events connecting the microbiota-gut-brain axis.

### 6.4. Fecal Microbiota Transplantation for Improving Mental Health via Epigenetic Changes

Fecal microbiota transplantation is considered a promising strategy for improving human mental health via modulation of the gut–brain axis. Fecal microbiota transplantation is defined as the transfer of intestinal microbiota from one individual to another and typically is carried out through colonoscopy [202]. Fecal microbiota transplantation is still in its infancy stage and should not be considered to be a treatment option outside the research setting owing to concerns regarding its long-term safety and efficacy [203]. It can be used to reshape the gut microbiome and investigate molecular mechanisms involved in the brain function in different diseases [204,205]. For example, Hu et al. constructed a mice model by fecal microbiota transplantation to probe the effects of gut microbiota in the neuropathogenesis (brain TRANK1 (Tetratricopeptide Repeat And Ankyrin Repeat Containing 1) expression and neuroinflammation) of bipolar disorder [206]. In another study, in a rodent model of autism, fecal microbiota transplantation or *Bifidobacterium* treatment could restore the fecal *Clostridium* spp. balance and rescue social interaction impairment and hippocampal BDNF expression [207].

Some current studies utilized this experimental procedure to induce mental diseases or improve mental health and prevent gut dysbiosis and anxiety-like behaviors [208]. For example, Giltay et al. reported that fecal microbiota transplantation could confer a decrease in anxiety symptoms and severity of depression in patents with *Clostridioides difficile* infections four weeks after fecal transplantation [209]. It has been found that the introduction of new species by allogenic fecal microbiota transplantation modulates the plasma metabolome and the epigenome. For instance, it was shown that the introduction of *Prevotella* ASVs is associated with DNA methylation alterations of actin filament-associated protein 1 (AFAP1) which affect mitochondrial functions and insulin sensitivity [210]. In another study, fecal microbiota transplant by gavage from autistic children to mice could confer the colonization of ASD-like microbiota and autistic behaviors associated with a decrease in DNA methylation of certain inflammatory genes [211]. More studies are required to confirm whether fecal microbiota transplantation can be used as a treatment option for improving mental health in humans by restoring epigenetic aberrations.

## 7. Conclusions and Future Prospects

The pathogenesis of many major mental disorders is associated with epigenetic aberrations. Currently, several lines of evidence suggest that disease pathogenesis can be linked to changes in the microbiome structure and composition via the microbiome–gut–brain axis. The complex interplay among environmental factors (in particular nutrition and maternal diet) and gastrointestinal microbiome consisting of more than three million functional genes, which interact with millions of human epigenetic marks and 30,000 human genes, and their role in accelerating or delaying the onset of mental disorders is a fundamental issue that has garnered much attention from the scientific community in recent years. Improved DNA/RNA sequencing technologies have paved the way for researchers to determine the exact strains of the affected bacteria and their interaction with genetic and epigenetic landscapes in disease pathogenesis. Describing microbial activity at the strain level will increase the chance of establishing a causal relationship among certain bacteria, epigenetic mechanisms, and specific mental disorders. More research on the modifications of the maternal microbiome during pregnancy by environmental factors and diet is still required to completely unravel the mechanisms of their roles in accelerating or delaying the onset of mental disorders via epigenetic changes. In particular, further research is still required to obtain greater insights on abnormally transmitted microbiomes on infant health, to mitigate the development of diseases such as ASD at early ages. Furthermore, comprehensive human studies with much larger sample sizes and well-balanced cohorts are necessary to confirm the potential benefits of probiotics, gut microbiome-derived metabolites, and different types of diets for epigenetic modulations alleviating major mental disorders.

## Figures and Tables

**Figure 1 genes-14-01506-f001:**
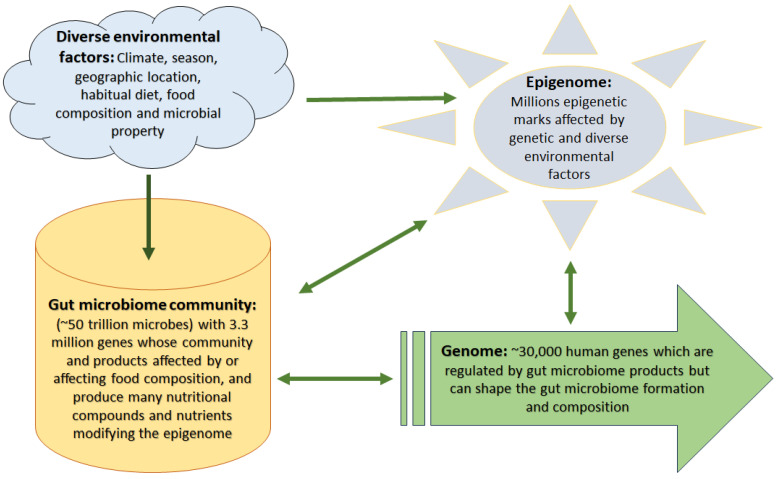
Complex interactions among environmental factors, gut microbiome, genome, and epigenome. Diverse environmental factors affect food quality and microbial habitat, thus influencing gut microbiome construction and composition that in turn alter the way they process food generating bioactive compounds and nutrients which affect the human epigenome and gene expression levels. However, external environmental factors and genetic construction can modify the epigenetic landscape as well, which in turn affects both gene expression levels and the gut microbiome community.

**Figure 2 genes-14-01506-f002:**
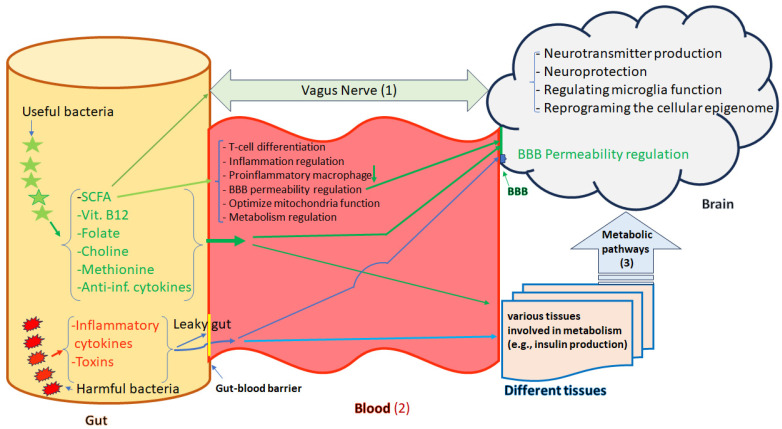
Microbiota-gut-brain axis. The gut and brain are linked together in three ways, (1) the Vagus nerve; (2) blood which is controlled by the BBB and (3) other tissues that produce varus metabolites and hormones (e.g., insulin). The Vagus nerve creates a direct reciprocal connection between the enteric nervous system (ENS) and the central nervous system (CNS). There are different types of bacteria in the gut either with useful or harmful effects which directly and indirectly may influence the brain functions. Bacterial fermentation metabolic byproducts such as short-chain fatty acids, vitamin B12, folate, choline, methionine, and anti-inflammatory cytokines have useful effects. These metabolic byproducts participate in immune modulation, BBB permeability regulation, optimization of mitochondrial function, neurotransmitter synthesis, neuroprotection, and reprogramming of the cellular epigenome. Harmful bacteria by producing toxins and inflammatory cytokines may cause dysbiosis and leaky gut increasing intestinal barrier permeability, which plays crucial roles in the pathophysiology of different diseases, including mental disorders by translocation of pro-inflammatory factors, chemokines, and pathogens into the blood stream and subsequently disrupting the BBB functions. Green Arrows address beneficial effects and bule arrows address harmful effects.

**Figure 3 genes-14-01506-f003:**
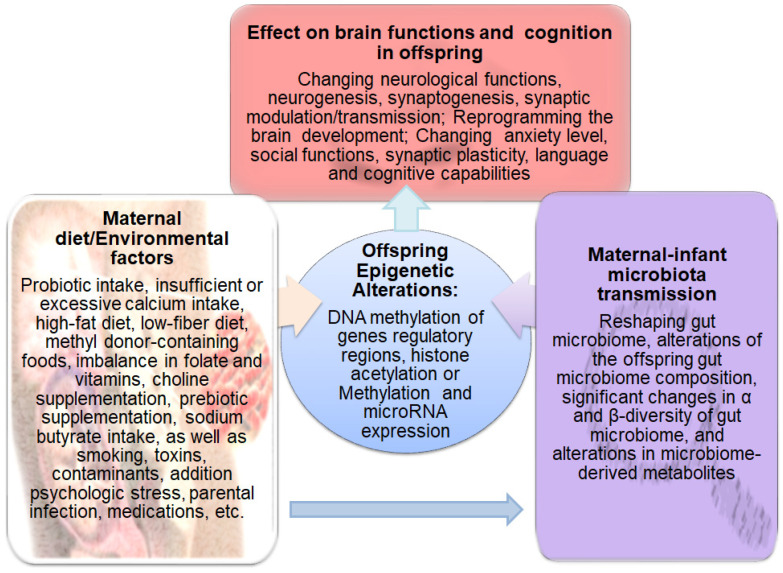
Schematic representation of the factors that affect the gut microbiome and epigenetic programing in offspring and subsequently influence brain functions and cognition. Diet and other environmental factors such as medications, toxins, contaminants, stress, smoking, addiction, and infection influence the maternal microbiome and epigenome, which in turn alter the offspring’s gut microbiome composition, microbiome-derived metabolites, and epigenetic landscape. These alterations affect brain functions and cognition leading to the development of mental diseases in offspring.

**Figure 4 genes-14-01506-f004:**
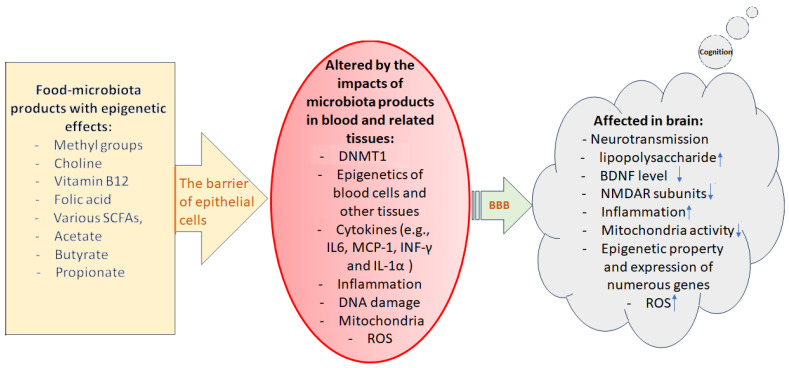
Illustration of the cascade of events during which epigenetically active food-microbiome products finally affect the brain’s molecular events and function, thus cognition. Epithelial cells of the intestinal wall are the first barriers determining which gut compositions can pass through blood circulation. In the next step, blood–brain barrier (BBB) defines what can pass through endothelial cells reaching to brain cells/tissue. A leaky gut or affected BBB may allow unhealthy materials to penetrate through these very important barriers altering the functional status of the affected cells/tissues. Downward arrows indicate decrease and upward arrows indicate increase.

**Table 1 genes-14-01506-t001:** Epigenetic aberrations associated with gut microbiota alterations in major mental diseases.

Mental Disorders	Altered Gut Microbiota or Its Products	Effects or Outcomes	Epigenetic Alterations	Ref.
Schizophrenia	Increased microbial diversity in schizophrenia patients	A link between schizophrenia, immunity, and microbial products in blood	DNA methylation	[93]
Schizophrenia	Butyric acid as a microbiome-derived metabolite	Increasing serum levels of butyric acid as a microbiome-derived metabolite in schizophrenia patients after 24-week risperidone treatment	Histone acetylation	[94]
Schizophrenia	Metabolic alterations in the gut microbiome	Abnormal short-chain fatty acid-producing bacteria in patients	Histone acetylation	[95]
Depression	Changes in microbiome composition	Higher plasma concentrations of pro-inflammatory cytokines (IL-8 and TNF-α) and differential DNA methylation at immune-metabolic genes in monocytes	DNA methylation	[96]
MDD	Possible role of microbiome	-----------	DNA methylation	[97]
MDD	Gut microbiota dysbiosis and pathogenesis of MDD	- Identifying 986 lysine acetylation sites in 543 proteins - Close association between lysine acetylation alterations and mitochondrial dysfunction in the brain	Lysine acetylation	[98]
Polycystic ovarian syndrome-associated depression	Significant differences in bacterial diversity and community, and stress responses in patients vs. healthy group	Lower FK506-binding protein 5 (FKBP5) DNA methylation in PCOS-associated depression	DNA methylation (reduced)	[99]
Depression	Gut microbiota dysbiosis-induced depression in Mice	Altered expression of 624 succinylation sites on 494 proteins and 315 acetylation sites on 223 proteins in gut microbiota dysbiosis	Acetylation and succinylation of proteins	[100]
ASD	Autism-gut microbiome associations	Microbiome differences in ASD may reflect dietary preferences	DNA methylation	[101]
ASD	Impairment in microbiome-related metabolites	Reducing protein and DNA methylation in autistic children, with concomitant lower concentrations of vitamins B6, B9 and B12	DNA methylation (reduced)	[102]
ASD	Changes in microbiome-related metabolites	Increased abundance of valeric acid-associated bacteria (*Acidobacteria*) and decreased abundances of key butyrate-producing taxa (*Ruminococcaceae, Eubacterium, Lachnospiraceae,* and *Erysipelotrichaceae*) in autistics	Histone acetylation	[103]
ASD	Compositional changes in the gut microbiome and its secondary metabolites	Decreased abundances of *Faecalibacterium* and *Agathobacter* and reduced 3-hydroxybutyric acid and melatonin levels in ASD children with a sleep disorder	Histone acetylation	[104]
ASD	Association between microbiome composition and dysregulated immune profiles	Altered hematopoiesis during embryogenesis and reduced expression of AP-1 complex for microglia development via dysregulation of HDAC1-mediated epigenetic machinery	Histone acetylation	[105]
Bipolar disorder	Alterations in the gut microbiome diversity	Negative correlation between CpG methylation status of the clock gene ARNTL and gut microbiome diversity	DNA methylation	[106]

**Table 2 genes-14-01506-t002:** External/environmental factors that affect the gut microbiome, brain functions, and epigenome.

Factors	Effects on Gut Microbiome	Effects on Brain Functions	Epigenetic Changes	Ref.
Probiotic intake (feeding of a probiotic organism, *Lactobacillus reuteri* to pregnant animals)	Maternal gut microbiome alterations	Changing neurological functions (neurogenesis synaptogenesis, and synaptic modulation/transmission)	Differentially methylated genes in FXS-like mice descended from mothers treated and non-treated with *L. reuteri*	[125]
Dietary insufficient or excessive calcium intake during pregnancy	Long-lasting adverse effects on the gut microbiome (unpublished)	Detrimental effects on brain development and function	Hypomethylation of Fads2 promoter in the brain of 21-day-old offspring in the group with reproductive diet, with low calcium concentration (LC 0.25%)	[126]
Maternal Stress	Alterations in the vaginal microbiome	Loss of maternal vaginal Lactobacillus leads to reduced transmission of this bacterium to offspring, altering reprogramming of the developing brain	Changes in genomic DNA	[127]
Maternal high-fat diet (HFD)	Alterations in the offspring gut microbiome	Predisposition to an ASD-like phenotype in male adolescent offspring	Enhanced cortical global DNA methylation and the expression of miR-423 and miR-494	[128]
Maternal HFD (>60% calories from fat)	Alterations in the offspring gut microbiome	Close association between maternal HFD with transgenerational susceptibility to chronic anxiety and alcohol abuse	DNA methylation changes (5 mC/5 hmC) in the genome regulatory regions	[129]
Maternal HFD	Alterations in the offspring gut microbiome	Social dysfunction and deficits in synaptic plasticity deficits in male offspring	Changes in histone acetylation	[130]
Methyl donor-containing foods in the maternal diet such as betaine compound	Alterations in the offspring gut microbiome	Direct association between a prolonged period of postnatal maturation of the prefrontal cortex and increased DNA methylation over time	DNA methylation	[131]
Imbalance in folate and vitamin B12 in maternal/parental diet	Alterations in gut microbiome composition	Folate and vitamin B12 are master regulators of brain DNA methylation	Changing global DNA methylation in the brain	[132]
Maternal choline supplementation and high-fat feeding	Alterations of gut microbiome composition	Potent modifier of brain DNA methylation	Brain DNA methylation	[133]
Prenatal chlorpyrifos exposure	Alterations in gut microbiome composition	Changing the cognitive and language domains	Increasing PPARγ DNA methylation	[134,135]
Maternal prebiotic supplementation (maternal galacto-oligosaccharide intake)	Increasing fecal microbiome-derived metabolites (butyrate and propionate)	- Altering brain and behavior in naïve and endotoxin-challenged offspring, increasing social preference and reducing anxiety	Increasing histone acetylation	[136]
Maternal low-fiber diet (MLFD)	Altering microbiome-derived metabolites, mostly butyrate	Impairment of cognitive function and synaptic plasticity in offspring	Reducing histone acetylation	[137]
High-dietary fiber intake	Reshaping gut microbiome	Reducing maternal obesity-induced cognitive and social dysfunctions	Changing histone acetylation	[138]
Maternal sodium butyrate intake	Altering the gut microbial metabolite butyrate	- Preventing long-term synaptic plasticity deficits, cerebellar cortex hypertrophy, and Purkinje cells firing - Improving ASD-like symptoms in offspring	Changing histone acetylation	[139]
Maternal smoking	Alterations of gut microbiome composition	Developing mental health diseases in offspring	Changing DNA methylation patterns	[140]
Maternal folate deficiency	Alterations of gut microbiome composition	Increasing the risk of neural tube defects (NTDs)	-Decreasing nuclear acetyl CoA levels and consequently reducing histone acetylation -Increasing lysine crotonylome as an epigenetic mark	[141]
Parental infection	Alterations in the offspring gut microbiome	Epigenetic alterations in the developing brain	Changes in germline epigenetics (expression of DNA methyltransferases and histone deacetylases)	[142]
Maternal intake of sulforaphane glucosinolate	Alterations in gut microbiome α and β-diversity in 3-week-old offspring	Reducing stress-related psychiatric diseases in offspring	Possible role of epigenetic modifications by sulforaphane glucosinolate	[143]

## Data Availability

Not applicable.

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
