# Peer review of "Epigenetic Aberrations in Major Psychiatric Diseases Related to Diet and Gut Microbiome Alterations"

_genes, 2023, doi:10.3390/genes14071506_

Round 1

Reviewer 1 Report

The authors describe in detail the effect of diet and gut microbiome on psychiatric diseases. The review is well structured and clearly explain the epigenetic modifications due to diet or gut microbiome. Minor revision should be addressed:

In the first sentence of the abstract you write: “Nutrition and metabolism play a powerful role in epigenetic modifications like histone acetylation and DNA methylation”, histone acetylation and DNA methylation are epigenetic signatures; so I suggest modifying the statement as follows: “…epigenetic modifications affecting histone acetylation and DNA methylation” or “Nutrition and metabolism modify epigenetic signatures like histone acetylation and DNA methylation”.

In line 45 the word as is missing: “…environmental factors such as toxins…”

In line 112 I suggest changing the order of the words as follows: “We will also provide…”

In table 1 I suggest removing “the type of” and directly write: Mental disorders (1st column) and Epigenetic alterations (4th column).

In line 274 there’s something missing because you just write: “Figure 2 illustrate”

In line 358 please remove while.

Author Response

Reviewer 1#

We thank you for your valuable comments. We carefully revised the manuscript based on your comments. The recommended modifications are made, and more information were added to the manuscript, all highlighted in yellow.

Comments and Suggestions for Authors

The authors describe in detail the effect of diet and gut microbiome on psychiatric diseases. The review is well structured and clearly explain the epigenetic modifications due to diet or gut microbiome. Minor revision should be addressed:

In the first sentence of the abstract you write: “Nutrition and metabolism play a powerful role in epigenetic modifications like histone acetylation and DNA methylation”, histone acetylation and DNA methylation are epigenetic signatures; so I suggest modifying the statement as follows: “…epigenetic modifications affecting histone acetylation and DNA methylation” or “Nutrition and metabolism modify epigenetic signatures like histone acetylation and DNA methylation”.

Answer to the comment: We implemented the comment in the revised version (page 1, lines 9-10).

In line 45 the word as is missing: “…environmental factors such as toxins…”

Answer to comment: We revised the sentence based on your comment (now page 2, line 46).

In line 112 I suggest changing the order of the words as follows: “We will also provide…”

Answer to comment: We revised the sentence based on your comment (now page 3, line 100).

In table 1 I suggest removing “the type of” and directly write: Mental disorders (1st column) and Epigenetic alterations (4th column).

Answer to the comment: Table 1 has been revised as recommended.

In line 274 there’s something missing because you just write: “Figure 2 illustrate”

Answer to comment: We checked the sentence again and revised it. Figure 3 illustrates the maternal diet and environmental factors that affect gut microbiome and epigenetic programing in offspring and subsequently influencing brain functions and cognition (Page 9, lines 297-299).

In line 358 please remove while.

Answer to comment: This word has been removed.

Reviewer 2 Report

Work by Nohesar et al. concerning Epigenetic aberrations in major psychiatric diseases related to diet and gut microbiome alterations is an interesting literature item, however, requiring several significant changes:

- contact and affiliation details of all co-authors of the manuscript are missing;

- citations throughout the text, in accordance with the journal's requirements, should be placed in the text in square brackets, please correct them;

- the introduction is too extensive, please edit and shorten it, some of the more detailed information should already be included in the following subsections;

- for a better presentation of the discussed content and due to the review nature of the work, please add a figure showing the Microbiome-Gut-Brain Axis dependencies/interactions to chapter 3;

- I am asking you to change figure 2, because in the current version it is very hard to read;

-please ensure that all names of microorganisms are written in italics in the text (this is missing in section 6.3) and that all abbreviations used are expanded first (e.g. TNF-α, IL-6, DNMT1 etc.)

- please also complete Author Contributions and reformat References in accordance with the requirements of the journal

Author Response

Reviewer 2#

The authors highly appreciate your attention to the errors of this manuscript and revised the manuscript based on your comments. The appropriate modifications are made, and more information were added to the manuscript, all highlighted in green.

Comments:

Comments and Suggestions for Authors

Work by Nohesar et al. concerning Epigenetic aberrations in major psychiatric diseases related to diet and gut microbiome alterations is an interesting literature item, however, requiring several significant changes:

- contact and affiliation details of all co-authors of the manuscript are missing;

Answer to the comment: We added information about contact and affiliation details of all co-authors of the manuscript.

- Citations throughout the text, in accordance with the journal's requirements, should be placed in the text in square brackets, please correct them;

Answer to the comment: We organized references in accordance with the journal's requirements.

- The introduction is too extensive, please edit and shorten it, some of the more detailed information should already be included in the following subsections;

Answer to the comment: Almost 200 words were removed from the introduction section.

- for a better presentation of the discussed content and due to the review nature of the work, please add a figure showing the Microbiome-Gut-Brain Axis dependencies/interactions to chapter 3;

Answer to the comment: We added a new figure to this section (Fig. 2).

- I am asking you to change figure 2, because in the current version it is very hard to read;

Answer to the comment: Thanks for the feedback. We revised figure 2 (now numbered to Figure 3).

-please ensure that all names of microorganisms are written in italics in the text (this is missing in section 6.3) and that all abbreviations used are expanded first (e.g. TNF-α, IL-6, DNMT1 etc.)

 Answer to comment: We corrected all errors related to this comment.

- please also complete Author Contributions and reformat References in accordance with the requirements of the journal

Answer to the comment: We completed Author Contributions and reformatted References in accordance with the requirements of the journal.

Reviewer 3 Report

The author highlights the significant role of nutrition, metabolism, and the gut microbiome in influencing epigenetic modifications in the brain. The authors further explore the connection between genetic makeup, gut microbiome composition, nutrition, and epigenetic changes in mental disorders. They also discuss the association between the gut microbiome, oxidative stress, inflammation, and mental health and examine the potential of probiotics to influence mental health through epigenetic modifications.

The manuscript is well-written and clear, although it contains a few mistakes, such as in line 61 or 490. Nonetheless, the manuscript overall provides a clear understanding of the topic. I suggest concluding each paragraph with a summary. Moreover,  I recommend delving into and commenting on the potential role of fecal transplants as a therapeutic approach, explicitly exploring how this method can influence and modulate epigenetic mechanisms.

Author Response

Reviewer 3#

We highly appreciate your feedback on our manuscript. We have revised the manuscript based on your valuable comments. The recommended modifications are made, and more information are added to the manuscript, all highlights in purple.  

Comments and Suggestions for Authors

The author highlights the significant role of nutrition, metabolism, and the gut microbiome in influencing epigenetic modifications in the brain. The authors further explore the connection between genetic makeup, gut microbiome composition, nutrition, and epigenetic changes in mental disorders. They also discuss the association between the gut microbiome, oxidative stress, inflammation, and mental health and examine the potential of probiotics to influence mental health through epigenetic modifications.

The manuscript is well-written and clear, although it contains a few mistakes, such as in line 61 or 490.

Answer to the comment: Thanks for this precise comment and sorry for our mistakes. We checked the text another time and corrected errors (now Page 2, lines 59-60).  

Nonetheless, the manuscript overall provides a clear understanding of the topic. I suggest concluding each paragraph with a summary.

Answer to the comment: Thanks for this valuable suggestion. We added a short conclusion to each section.

 Moreover,  I recommend delving into and commenting on the potential role of fecal transplants as a therapeutic approach, explicitly exploring how this method can influence and modulate epigenetic mechanisms.

Answer to the comment: Based on your feedback, we added a new section named “6.4. Fecal microbiota transplantation for improving mental health via epigenetic changes” (Page 15-16, lines 517-546).

Round 2

Reviewer 2 Report

Thank you very much for all the changes you made, now your manuscript is much more readable. Good luck with your further work!